# Binary Expression Enhances Reliability of Messaging in Gene Networks

**DOI:** 10.3390/e22040479

**Published:** 2020-04-22

**Authors:** Leonardo R. Gama, Guilherme Giovanini, Gábor Balázsi, Alexandre F. Ramos

**Affiliations:** 1Departamento de Radiologia e Oncologia & Instituto do Câncer do Estado de São Paulo—Faculdade de Medicina, Universidade de São Paulo, São Paulo CEP 05403-911, SP, Brazil; 2Escola de Artes, Ciências e Humanidades, Universidade de São Paulo, Av. Arlindo Béttio, 1000, São Paulo CEP 03828-000, SP, Brazil; 3The Louis and Beatrice Laufer Center for Physical and Quantitative Biology, Stony Brook University, Stony Brook, NY 11794, USA; 4Department of Biomedical Engineering, Stony Brook University, Stony Brook, NY 11794, USA

**Keywords:** noise in gene networks, binary gene regulation, bursty gene expression, Shannon theory, stochastic modeling

## Abstract

The promoter state of a gene and its expression levels are modulated by the amounts of transcription factors interacting with its regulatory regions. Hence, one may interpret a gene network as a communicating system in which the state of the promoter of a gene (the *source*) is communicated by the amounts of transcription factors that it expresses (the *message*) to modulate the state of the promoter and expression levels of another gene (the *receptor*). The reliability of the gene network dynamics can be quantified by Shannon’s entropy of the message and the mutual information between the message and the promoter state. Here we consider a stochastic model for a binary gene and use its exact steady state solutions to calculate the entropy and mutual information. We show that a slow switching promoter with long and equally standing ON and OFF states maximizes the mutual information and reduces entropy. That is a binary gene expression regime generating a high variance message governed by a bimodal probability distribution with peaks of the same height. Our results indicate that Shannon’s theory can be a powerful framework for understanding how bursty gene expression conciliates with the striking spatio-temporal precision exhibited in pattern formation of developing organisms.

## 1. Introduction

The remarkable spatio-temporal precision observed during development of *metazoa* is the result of well-orchestrated gene networks regulating the expression of proteins that determine cell fate. That contrasts with the unavoidable fluctuations of intracellular environment caused by low copy numbers of reactants [1] as detected in expression of genes of prokaryotic [2] and eukaryotic cells [3] and high precision experiments demonstrating the pulsed transcription of developmental genes of social amoeba *Dyctiostelium* [4], stochastic gene activation [5] and transcriptional bursting [6,7] in *Drosophila* embryos, or the bursty transcription of mammal genes [8]. Such features may lead one to ask whether in the due course of evolution there emerged mechanisms equipping biological systems with the capacity of modulating noise to exploit it functionally [9,10,11,12]. One example is the ubiquitous bursty gene expression [13,14] taking place during *Drosophila* embryogenesis when the gene networks govern multiple proteins to have proper amounts within proper spatial and temporal ranges during pattern formation [15,16,17]. For such patterns to be achieved, the activation and inactivation of the promoters of the genes of developmental networks should be modulated with sufficient precision. Hence, one may ask how the promoters, and transcription factors regulating the expression of these gene networks, contribute to the emergence of such a reliable dynamics.

Here we approach those questions considering a gene network as a communicating system in which a gene (the source) expresses transcription factors (the message) that regulate another gene (the receptor). That enables the application of information theory for investigating the transduction of an input signal into a functional output by a gene network [18,19,20]. As opposed to the common consideration of gene activation/inactivation as completely random events, here we assign to them information content originating, for example, from upstream regulators. Data for nascent transcripts in *Drosophila* embryos indicates that the position of activated promoters is in good agreement with that of their corresponding proteins during embryogenesis [13,14,21,22]. Therefore, we assume the information content of the input message to be the promoter state as the source, since the binding and unbinding of transcription factors to the enhancer induces the state of the promoter of the receptor gene [23,24]. The processing of all information arriving at the receptor is not considered here [25,26,27,28], as we are interested on the reliability of the information content of the message.

Shannon’s information theory is employed to compute the entropy of the message and the mutual information between the message and the promoter state of the source [29]. Previous applications of information theory to biological problems employed a parameter-less analysis of experimental data to investigate optimal limits of information flow [19,30,31]. The absence of an underlying effective model, however, makes it difficult to biologically interpret the obtained results because of a lack of a mechanistic description of how optimal or near optimal information flow is achieved. The application of information theory to an exactly solvable stochastic model with a simple biological interpretation might help to fulfill that lacuna. Such an analysis may provide good guidance to synthetic biologists on designing gene circuits to behave as reliable (or unreliable) messaging devices. It can also be used as a basic building block useful to understand biological phenomena being described within the framework of information transmission [32,33], such as embryogenesis [34,35,36], carcinogenesis [37,38], or cell phenotype adaptation to fluctuating environments [39,40,41].

We investigate how unavoidable intracellular noise can be exploited by living organisms to produce reliable information exchange. We use the exact solutions of the stochastic binary model for gene expression formulated for coupled transcription-translation [42,43]. Although the binary model was later enhanced to describe transcription and translation as separate processes [44,45,46], its preliminary interpretation is sufficient to provide useful insights on the workings of a gene network. Our approximation implies, considering that, e.g., post-transcriptional regulation is absent, that translation delay is short in comparison with the relevant time scales governing the whole protein synthesis process, and that there is an approximately linear relation between the mRNA and protein numbers [47]. Experimental data for *Drosophila* embryos showing a good agreement between spatial patterns of transcripts and transcription factors [13,21] justify our approximation.

The remaining parts of this manuscript are organized as follows. Section 2 presents our picture of a gene network as a communicating system, and the equations used to compute the probabilities, noise, average values, entropy and mutual information for the binary gene. Our results are presented at Section 3 as a set of graphs relating the entropy and mutual information to gene expression noise, promoter switching speeds, average number of transcription factors and their corresponding probability distributions. A discussion of how our results relate to a non-exhaustive set of previous experimental or theoretical analysis related with our investigation is presented at Section 4. The results and analysis presented here contribute to the understanding of the detrimental role of noise for the proper development of an organism, or its benefits for the adaptation and robustness of living systems challenged by an ever changing environment.

## 2. Model and Methods

To set a reference value for a comparative analysis, we will evaluate the entropy of the message sent by a constitutive and by a binary gene at the steady state regime. The promoter of the constitutive gene is always ON while it switches between ON and OFF states on the latter [48]. We assume coupled transcription-translation for a clearer interpretation of our results. Our analysis focus on understanding: *i.* the entropy of a message sent by both the constitutive and binary sources; *ii.* the mutual information between the promoter state and the number of transcription factors sent by the binary source. These enable the investigation of the conditions for a binary source to send a reliable or non-reliable message about its state being ON or OFF. In this section, we present the formulas used to perform our calculation and their biological interpretation. The codes for performing the numerical calculations presented in this manuscript are available online (https://github.com/amphybio/stochastic-gene-expression).

### 2.1. Qualitative Model and Noise Quantification

Figure 1 shows the effective set of reactions that we used to interpret a gene network as a communicating system. It depicts the symbols for: synthesis rate of the transcription factors when the promoter is ON, *k*; transcription factor degradation rate, ρ; promoter switching rate from ON to OFF state and vice-versa, *h* and *f*, respectively; the number of transcription factors available, *n*. No synthesis happens when the promoter is OFF. The curved lines indicate the interaction of the protein encoded in one gene with the enhancer of the receptor.

The number of transcription factors is a random variable and the noise of the message can be computed using the Fano factor, defined as the ratio between the variance of *n* to its average value. The average number of transcription factors (or expected message from the source) is denoted by 〈n〉, and the Fano factor by F, such that:(1)$$F=\frac{\langle n^{2}\rangle -{\langle n\rangle }^{2}}{\langle n\rangle }\;\;.$$

The Fano factor is a measure of how similar to Poissonian a probability distribution is. The Poisson distribution has F=1 while F<1 characterizes a sub-Poisson distribution. A super-Poissonian distribution has F>1.

### 2.2. A Stochastic Model for a Constitutive Gene

We model a constitutive gene as a Poissonian birth and death process. The random variable *n* indicates the number of transcription factors available for regulating a receptor gene. The probability of finding *n* transcription factors is indicated by $\phi_{n}$. The synthesis and degradation rates are denoted by *k* and ρ, respectively. The probability $\phi_{n}$ is given by
(2)$$\phi_{n}=\frac{e^{-N}}{n!}N^{n}\;\;,$$
where
(3)$$N=\frac{k}{\rho }\;\;,$$
is the average number of transcription factors produced from a constitutive gene at the steady state regime. The variance of the Poissonian process is also *N*.

### 2.3. A Stochastic Model for the Binary Gene

We model the binary gene as two Poissonian birth and death processes coupled by a random telegraph process. The random telegraph process describes the switching of the promoter between ON and OFF states. When the promoter is ON (or OFF) proteins are synthesized at a rate *k* (or 0) and degradation happens at rate ρ. There are two random variables, (m,n), where *m* indicates the state of the promoter as ON or OFF and *n* the number of proteins. The probability of finding the promoter ON (or OFF) and *n* proteins is indicated by $\alpha_{n}$ (or $\beta_{n}$). Using the exact solutions for this model [42,43] we have: (4)$$\begin{array}{ccc} \alpha_{n} & = & p_{\alpha }\frac{N^{n}}{n!}\frac{{\left(1+\epsilon p_{\alpha }\right)}_{n}}{{\left(1+\epsilon \right)}_{n}}M\left(1+\epsilon p_{\alpha }+n,1+\epsilon +n,-N\right), \end{array}$$(5)$$\begin{array}{ccc} \beta_{n} & = & \left(1-p_{\alpha }\right)\frac{N^{n}}{n!}\frac{{\left(\epsilon p_{\alpha }\right)}_{n}}{{\left(1+\epsilon \right)}_{n}}M\left(\epsilon p_{\alpha }+n,1+\epsilon +n,-N\right), \end{array}$$(6)$$\begin{array}{ccc} \phi_{n} & = & \frac{N^{n}}{n!}\frac{{\left(\epsilon p_{\alpha }\right)}_{n}}{{\left(\epsilon \right)}_{n}}M\left(\epsilon p_{\alpha }+n,\epsilon +n,-N\right), \end{array}$$
where the constants $p_{\alpha }$, ϵ and *N* are defined as:(7)$$p_{\alpha }=\frac{f}{f+h}\;\;,p_{\beta }=\frac{h}{f+h}\;\;,\;\;\epsilon =\frac{f+h}{\rho }\;\;,\;\;N=\frac{k}{\rho }\;\;.$$
$\phi_{n}$ indicates the marginal probability of finding *n* proteins, namely $\phi_{n}=\alpha_{n}+\beta_{n}$, and M(a,b,x) denotes the KummerM function, defined as a power series:$$M\left(a,b,x\right)=\sum_{n=0}^{\infty }\frac{{\left(a\right)}_{n}}{{\left(b\right)}_{n}}\frac{x^{n}}{n!}\;\;,$$
where ${\left(a\right)}_{n}=a\left(a+1\right)\ldots \left(a+n-1\right)$ denotes the Pochhammer symbol [49].

#### 2.3.1. Interpretation of the Parameters $P_{\alpha }$, ϵ, and *N*.

The parameters $p_{\alpha }$ and $p_{\beta }$ are the steady state marginal probabilities for the promoter to be at the ON and OFF states, namely $p_{\alpha }=\sum_{n=0}^{\infty }\alpha_{n}$ and $p_{\beta }=\sum_{n=0}^{\infty }\beta_{n}$ with $p_{\alpha }+p_{\beta }=1$. $p_{\alpha }$ (or $p_{\beta }$) give the average fraction of a time interval *T* that the promoter will be ON (or OFF) while switching at random between these states.

For the promoter being exclusively ON the source would become constitutive (see Section 2.2) with average number of transcription factors being *N* (see Equation (Equation 3) or Equation (Equation 7)).

The parameter ϵ is the ratio of the promoter switching rate to the transcription factors degradation rate. It can also be interpreted as the ratio of the transcription factor’s lifetime to the average period of a complete cycle of promoter switching (see Equation (Equation 7)) for a fixed $p_{\alpha }$. For simplicity, we may assume $T_{D}=\frac{1}{\rho }$ as the protein lifetime, and $T_{S}=\frac{f+h}{fh}$ as the average time for the promoter to complete one switching cycle. Using the definitions at Equation (Equation 7) we obtain
$$\epsilon =\frac{1}{p_{\alpha }p_{\beta }}\;\frac{T_{D}}{T_{S}}\;\;.$$

When ϵ≪1 the switching cycle is long and the protein’s lifetime is shorter than the time spent by the promoter at ON or OFF state if $p_{\alpha }∼1/2$. When ϵ≫1 the switching cycle is very short compared to the transcription factors lifetime and multiple switchings will happen while a transcription factor remains functional.

The parameter *N* is the average number of transcription factors if the binary gene remains ON all the time as if it was a constitutive gene.

#### 2.3.2. The Mean Number and the Conditional Mean Number of the Stochastic Model for a Binary Gene

The average number of transcription factors synthesized from the binary gene (〈n〉), and the average number of transcription factors given that the promoter is ON or OFF (denoted by $\langle n_{\alpha }\rangle$ or $\langle n_{\beta }\rangle$) are
(8)$$\langle n\rangle =Np_{\alpha },\;\;\;\langle n_{\alpha }\rangle =N\frac{1+p_{\alpha }\;\epsilon }{1+\epsilon },\;\;\;\langle n_{\beta }\rangle =N\frac{p_{\alpha }\;\epsilon }{1+\epsilon }\;\;.$$

The mean number of transcription factors of the binary gene depends on the proportion of time that the promoter of the gene is ON. Hence, the average amount of transcription factors will range from zero to *N* accordingly with $p_{\alpha }$. The conditional mean number of transcription factors is obtained using $\langle n_{\alpha }\rangle =\frac{1}{p_{\alpha }}\sum_{n=0}^{\infty }n\alpha_{n}$ and $\langle n_{\beta }\rangle =\frac{1}{p_{\beta }}\sum_{n=0}^{\infty }n\beta_{n}$ which can be obtained using the generating function technique [43,45]. The mean number of transcription factors can be written as $\langle n\rangle =p_{\alpha }\langle n_{\alpha }\rangle +p_{\beta }\langle n_{\beta }\rangle$.

For very small values of ϵ, $\langle n_{\alpha }\rangle \to N$ and $\langle n_{\beta }\rangle \to 0$. The binary gene behaves as a constitutive one when its promoter is ON and as a silenced one when its promoter is OFF. The average number is a quantity between these two values that does not provide a useful information about the instantaneous state of the system. However, measurement of the instantaneous number of transcription factors is useful to understand the promoter state.

For large values of ϵ we have $\langle n_{\alpha }\rangle ,\;\;\langle n_{\beta }\rangle ∼\langle n\rangle$. The state of the promoter is indistinguishable by simply measuring the number of transcription factors. At that limit, the source approaches the behavior of a constitutive one.

#### 2.3.3. The Variance and Bursting Noise of *N* on the Stochastic Model for a Binary Gene

The variance of *n*, denoted by $\sigma^{2}$, and the Fano factor (F) when the transcription factors are produced from a binary gene are, respectively,
(9)$$\sigma^{2}=\langle n\rangle \left(1+\frac{N\;(1-p_{\alpha })}{1+\epsilon }\right),\;\;\;F=1+\frac{N\;(1-p_{\alpha })}{1+\epsilon }=1+\delta \;\;.$$

Since the discussion about the noise is similar for both $\sigma^{2}$ and F we focus on the second. Because of the promoter switching, the value of F is larger than that of a constitutive one by a quantity $\delta =N\;\left(1-p_{\alpha }\right)/\left(1+\epsilon \right)$.

The Fano factor is computed using the marginal probability $\phi_{n}$, the probability of finding *n* transcription factors independently of the promoter state. The increment δ helps to understand the hyper-Poissonian nature of the message sent by the binary source. A Poissonian distribution describing the constitutive source has δ=0 while in a super-Poissonian distribution describing the message sent by the binary source we have δ>0. The super-Poissonian distribution describes multiple qualitatively distinct gene expression regimes.

For $p_{\alpha }=1$ the promoter is fully ON, the binary source becomes constitutive and δ=0. For $p_{\alpha }=0$ we have a fully repressed gene. Additionally, when ϵ approaches infinity the noise of the message sent by the binary gene is similar to that of a constitutive one if *N* and $p_{\alpha }$ are finite constants. For $\left(N,\;p_{\alpha }\right)$ being finite constants and ϵ approaching zero we have $\delta \to N(1-p_{\alpha })$.

For instance, a bursty expression characterized by k,h≫ρ and h≫f or N,ϵ≫1 and $p_{\alpha }\ll 1$ (see Equation (Equation 7)) has δ≈k/h or δ≈N/ϵ which is called the burst size. This is a mathematical entity with interpretation as transcriptional or translational dependent on the meaning of the random variable *n*, which in here denotes the number of transcription factors. Hence, we interpret this limit as the translational bursting and δ as the translational burst size. Note, however, that in literature it has also been interpreted as the transcriptional bursting size [44,46,50,51,52]. Here, the burst size δ∼k/h coincides with the noise increment in relation to the noise in the expression of the constitutive gene. In that case, the promoter remains OFF for a long interval, and switches ON for a brief time interval during which a burst takes place. Once the promoter switches OFF the amount of gene products decays exponentially until the next burst.

The binary expression, on the other hand, is characterized by f+h≪ρ or ϵ≪1. The number of synthesized transcription factors is governed by a bimodal distribution which peaks are centered around 0 and *N* weighted by the probabilities $p_{\beta }$ and $p_{\alpha }$, respectively. During steady state regime, the trajectories of the number of transcription factors governed by a bimodal distribution can be described approximately as follows. Initially, we set the promoter OFF and no transcription factors, then there will be, at random instants: an increase of *n* towards a finite value $\langle n_{\alpha }\rangle \approx N$ after the promoter switching ON; fluctuations of *n* around $\langle n_{\alpha }\rangle$ corresponding to a Poissonian process while the promoter remains ON; a sharp reduction towards zero after the promoter switching OFF; then n=0 while the promoter remains OFF; and the cycle repeats. Let us set $p_{\alpha }=1/2$ and take the limit of ϵ≪1 such that F→1+N/2. We set the promoter OFF and zero transcription factors at the initial instant. Once the promoter switches ON, *n* increases towards ∼N as if it were a time-dependent Poissonian process because the promoter state remains stable for a sufficiently long time interval. For the case of a fast growth of *n* in comparison with the remaining time scales of the system, one might take it as a burst which size would not correspond to the increment in noise, δ, but to 2δ, and an analysis of the full trajectory would be necessary to understand this noise regime [22,52].

The two examples above points to the necessity of a more precise classification of noise and its biological implications. Information theory constitutes a useful toolbox for that purpose.

### 2.4. Analyzing the Information Content of the Message

The information content of the message is quantified in the framework of Shannon’s communication theory, where the variance of binary signals correlates with their information content. Hence, the average information content of the message is given by Shannon’s entropy; the coupling between the message and the state of the promoter of the source is given by the mutual information; the average information content of the message given the promoter being ON is measured by the conditional entropy.

#### 2.4.1. Entropy for the Constitutive Source

The Shannon’s entropy of a probability distribution $\phi_{n}$ is denoted by H(n) and defined as $H\left(n\right)=-\sum_{n=0}^{\infty }\phi_{n}\log\phi_{n}$, with the logarithm taken to base 2. For a Poissonian distribution with average value *N* the entropy is given by
(10)$$H\left(n\right)=N-N\;\log\left(N\right)+e^{-N}\sum_{n=0}^{\infty }\frac{N^{n}}{n!}\;\log\left(n!\right)\;\;.$$

The entropy of the constitutive gene will be considered here as a reference value to be compared with that of the binary one.

#### 2.4.2. Entropy, Conditional Entropy and Mutual Information for the Binary Source

Since it is not trivial to find a closed form to compute the logarithm of a KummerM function, we obtained the entropy and mutual information for the binary gene by computing their formulas numerically. The entropy of the message sent by the binary gene is:(11)$$H\left(n\right)=-\sum_{n=0}^{\infty }\phi_{n}\log\phi_{n}\;\;,$$
where $\phi_{n}$ is given by Equation (Equation 6) and computed setting numerical values to its parameters given at Equation (Equation 7).

The conditional entropy of *n* given the promoter being ON or OFF is denoted by $H_{\alpha }\left(n\right)$ or $H_{\beta }\left(n\right)$, respectively, which are written as: (12)$$\begin{array}{ccc} H_{\alpha }\left(n\right) & = & \log\left(p_{\alpha }\right)-\frac{1}{p_{\alpha }}\sum_{n=0}^{\infty }\alpha_{n}\log\left(\alpha_{n}\right)\;\;, \end{array}$$(13)$$\begin{array}{ccc} H_{\beta }\left(n\right) & = & \log\left(p_{\beta }\right)-\frac{1}{p_{\beta }}\sum_{n=0}^{\infty }\beta_{n}\log\left(\beta_{n}\right)\;\;. \end{array}$$

These formulas are obtained from the definition of the conditional entropy of *X* given Y=y, namely $H\left(X|Y=y\right)=-\sum_{x\in X}p\left(X|Y=y\right)\log p(X|Y=y)$, where we assume *X* being the number of transcription factors and *Y* the promoter state. Here $\alpha_{n}$, $\beta_{n}$, and ($p_{\alpha }$, $p_{\beta }$) are, respectively, given by Equations (Equation 4), (Equation 5) and (Equation 7).

The mutual information between the message *n* and the source state being ON or OFF is given by
(14)$$I\left(n;m\right)=H\left(n\right)-p_{\alpha }\;H_{\alpha }\left(n\right)-p_{\beta }\;H_{\beta }\left(n\right)\;\;,$$
where H(n), $H_{\alpha }\left(n\right)$, and $H_{\beta }\left(n\right)$ are, respectively, given by Equations (Equation 11)–(Equation 13).

## 3. Results

We investigate the reliability of the information transfer considering the unavoidability of noise by characterizing the dynamics of the promoter and the distributions governing the message (gene product) generated by the source gene. Section 3.1 shows an analysis of the entropy and mutual information as a function of the Fano factor to motivate the hypothesis that noise properties are actively regulated by cells. Section 3.2 presents a characterization of the promoter switching dynamics to enable the condition of reduced entropy and increased mutual information of the message. Since the reliability of the signal is evaluated by means of the average signal, in Section 3.3 we investigate the behavior of the entropy and mutual information as functions of the average message. We also analyze the conditional entropy as a function of the average conditional message and determine the corresponding probability distributions for three distinctive switching dynamics: fast, slow, and intermediary.

Each subsection is structured with a set of graphs displaying the results obtained for a set of parameter values (see Equation (Equation 7)) indicated inside the graphs and on the figures’ titles. We also present a description of the graphs and the data obtained using the corresponding equations (see Section 2). A final paragraph is dedicated to discussing and interpret the results using the qualitative picture at Figure 1.

### 3.1. Binary Expression Enables Entropy Reduction and Mutual Information Increase

The horizontal axis of both graphs of Figure 2 shows the bursting quantified by the Fano factor (see Equation (Equation 1)). The vertical axis of graphs A and B shows, respectively, the entropy of the message (see Equation (Equation 11)) and the mutual information between the message and the state of the promoter of the source (see Equation (Equation 14)). Each continuous line is associated with a specific value of the promoter switching rate, ϵ (see Equation (Equation 7)). The dashed curves are isolines for fixed values of the probability for the promoter to be ON, $p_{\alpha }$. The color scheme for the curves is given at the caption. The black solid diamond gives the entropy of the message when it is sent by a Poissonian source which is usually employed on the description of a constitutive gene. Please note that we have no mutual information for the constitutive gene because we assume that its promoter will be always ON. A condition of equivalence between the binary gene to the constitutive one can be devised for ϵ>>N which corresponds to a switching cycle being much faster than the protein synthesis rate. In that case, the mutual information approaches zero.

*Graph A* shows that the constitutive gene sets a reference value for the entropy for a fixed 〈n〉. The binary source at the small noise regime (F∼1) generates a message which entropy is greater than that sent by the constitutive gene. As the noise increases, however, the entropy of the message sent by the binary source reaches a maximum and then starts decreasing. Then the entropy of the message sent by the binary source becomes smaller than that of the constitutive source and decays towards arbitrarily small values. Moreover, for a fixed value of F, the message with minimal entropy is generated by a process with the smallest ϵ.

*Graph B* shows that the mutual information between the message and the source state reaches a maximum for values of the Fano factor being ∼1+〈n〉 and ϵ≪1. For a fixed value of F the mutual information becomes minimal for the greatest values of ϵ. As the switching rate becomes slower, i.e., ϵ≪1, the mutual information is maximal for $p_{\alpha }∼1/2$. Furthermore, one may set an interval at which the mutual information has higher, significant values, by arbitrarily fixing the probability for the promoter to be ON within a range $1/L\leq p_{\alpha }\leq 1-1/L$ such that the corresponding noise is (L−1)〈n〉≥F≥〈n〉/(L−1), where L>1 and F≫1 such that (F−1)(1+ϵ)∼F.

The results presented here show that one may increase the expression noise of a binary gene to generate messages with arbitrarily small entropy. The reduction of the entropy and of its maximal value depends both on the message’s noise and on the probabilities governing the promoter states. Hence, reducing the entropy of the message is not sufficient for optimizing its reliability, so an analysis of the mutual information between the message and the promoter state is necessary. Indeed, the mutual information has a maximum when the probability for the promoter to be ON is 1/2 and the noise increment δ is approximately the average number of gene products. Therefore, besides promoter state probabilities, the noise in gene expression must also be regulated for the maximization of the information content of the message sent by the binary source.

### 3.2. The Slow Switching Genes Generate Reduced Entropy and Increased Values for Mutual Information

The horizontal axis of both graphs of Figure 3 indicates the values of the transcription factor’s lifetime relative to the ON and OFF switching cycle of the source’s promoter. The vertical axis of graphs A and B shows, respectively, the entropy of the message (see Equation (Equation 11)) and the mutual information between the message and the source state. To each continuous line we associate a specific value of the probability for the source to be ON, $p_{\alpha }$ (see Equation (Equation 7)). The dashed black curve indicates the entropy for the constitutive gene. This value is given as a reference since there is no switching on a constitutive promoter and, additionally, there is no mutual information between the promoter state and the message. The color scheme for the curves is given at the caption.

*Graph A* shows that the entropy for a given $p_{\alpha }$ is minimal when the switching is slow. As the switching rate increases, the entropy reaches a maximum value ($H_{m}$) for an $\epsilon_{m}$ which is greater as the probability for the promoter to be ON is reduced. For $\epsilon >\epsilon_{m}$ and a given $p_{\alpha }$, the entropy of the message decreases and asymptotically approaches that of a constitutive source. A source with ϵ≪1 generates a message for which the entropy increases with $p_{\alpha }$ while the opposite happens for large values of ϵ. Furthermore, for ϵ≪1 and $p_{\alpha }=1/2$ the entropy is smaller than that of the constitutive source, but not minimal.

*Graph B* shows the mutual information as a function of ϵ with each line being related to a specific value of $p_{\alpha }$. The similarity of the mutual information for complementary values of $p_{\alpha }$ as the switching rate increases relates to the bell-shaped curves (in semi-log scale) shown on Figure 2 (graph B). For a given value of $p_{\alpha }$ the mutual information decreases monotonically with the increase of ϵ. For ϵ≪1 the mutual information is maximal when $p_{\alpha }=1/2$, and for $p_{\alpha }\ll 1$ the message will have minimal entropy and minimal mutual information with the promoter state.

Equation (Equation 9) shows that the increment δ on the noise of the message sent by the binary gene can be regulated by three distinct parameters. The value of *N* is determined by the capacity of the cell of synthesizing transcription factors while $p_{\alpha }$ is given by the enhancer’s state. Therefore, minimizing ϵ may be a useful method for increasing δ. Indeed, here we have obtained reduced entropy and maximal mutual information for the slow switching genes. As expected, a very fast switching gene will generate a message with entropy approaching that of the constitutive gene with the same average expression levels. An additional regime is the bursting that takes place when we have ϵ,N≫1 and $p_{\alpha }\ll 1$ [52]. This regime leads to a strong reduction on the entropy of the message but its mutual information is low. Hence, it would not be reliable in the context of a gene network in which the information content of the message is the promoter state.

### 3.3. The Distributions of the Slow Switching Bursty Regime
with Reduced Entropy and Maximal Mutual Information Are
Bimodal

Figure 4, Figure 5 and Figure 6 have four graphs each where: graph A shows the entropy on vertical axis and the average number of gene products (〈n〉) on the horizontal; graph B shows the entropy conditional to the promoter being ON versus $\langle n_{\alpha }\rangle$ (see Equation (Equation 8)); graph C shows the mutual information versus 〈n〉; graph D shows the probability distribution of *n*. Each continuous line corresponds to a given $p_{\alpha }$ with color code given at the key of *graph B*. The values of the entropy (vertical axis of *graph A*); entropy conditional to the source being ON (vertical axis of *graph B*); mutual information (vertical axis of *graph C*) are obtained fixing a given $p_{\alpha }$ and varying *N* to generate the values of 〈n〉 at the horizontal axis. The probability distributions of *graph D* are obtained by also fixing 〈n〉=50, such that $N=50/p_{\alpha }$. For each $p_{\alpha }$ we compute the values of $\phi_{n}$ (continuous lines) and $\alpha_{n}$ (dashed lines). In each set of graphs, the value of ϵ is fixed and given at graph A’s title. Results for the constitutive gene are shown by a black line on graphs A, B, and D.

Figure 4. *Graph A* shows that the entropy of the message increases monotonically with 〈n〉 for both the constitutive and binary sources. As $p_{\alpha }$ approaches one the entropy of the binary sources approaches that of a constitutive. A source with smaller values of $p_{\alpha }$ will send messages with significantly reduced entropy which growth with 〈n〉 is much slower. *Graph B* shows the entropy of the message under the condition of the source being ON. In that case the conditional entropy of all messages approaches that of a constitutive source for all values of $p_{\alpha }$. That is observed by introducing the entropy for the constitutive source as a reference value (black curve). *Graph C* shows that the mutual information between the source’s promoter state and the message increases monotonically until it reaches a limit at a value that depends on $p_{\alpha }$. The mutual information for two complementary values of $p_{\alpha }$ approximate each other for a sufficiently large value of 〈n〉. For $p_{\alpha }=1/2$ the mutual information reaches its maximum with its plateau being reached when 〈n〉∼30. *Graph D* shows the probability distribution for 〈n〉=50. The black curve shows the probability distribution of *n* when emitted by a constitutive source. The distributions generated by the binary source are all bimodal, with one mode centered around zero and the other around *N* (where $N=\langle n\rangle /p_{\alpha }$). As the value of $p_{\alpha }$ approaches zero the second peak has its height lowered and its center displaced to the right. The opposite happens when $p_{\alpha }$ approximates unity with the taller peak approaching that of the constitutive source for the same mean.

Figure 5. *Graph A* shows that the entropy of the message increases monotonically with 〈n〉 for both the constitutive and the binary sources. For larger values of $p_{\alpha }$, the entropies of the binary sources are greater than that of a constitutive. Entropy of the message from the binary source is a bit smaller than that of a constitutive for small values of $p_{\alpha }$. *Graph B* shows the entropy of the message given the promoter of the source being ON. In that case the conditional entropy of all messages is greater than that of a constitutive source. *Graph C* shows that the mutual information also increases monotonically until it reaches a limit value that depends on $p_{\alpha }$. The mutual information for two complementary values of $p_{\alpha }$ approximate each other for a sufficiently large value of 〈n〉. For $p_{\alpha }=1/2$ the mutual information reaches its maximum with its plateau being reached when 〈n〉∼30. Note, however, that the maximal value for the mutual information is smaller than that for ϵ=0.01. *Graph D* shows the probability distributions for the binary source. The distributions generated by the binary source have significant probability values spread along a large range of values of *n*. For $p_{\alpha }=1/2$ the distribution is table shaped. As $p_{\alpha }\to 1$ the distribution forms a peak that gets closer to that of the distribution of a constitutive source while the distributions have a sector governed by a power law when $p_{\alpha }$ approaches zero.

Figure 6. *Graph A* shows that the entropy of the message increases monotonically with 〈n〉 for both the constitutive and the binary sources. The entropy of the message emitted by the binary source is larger than that of a constitutive gene for all values of $p_{\alpha }$. *Graph B* shows the entropy of the message given the promoter being ON. The conditional entropies of all messages are greater than that of a constitutive source. *Graph C* shows that the mutual information also increases monotonically until it reaches a limit value that depends on $p_{\alpha }$ and that this maximal value is smaller than that for ϵ=0.01 or ϵ=2. *Graph D* shows the probability distributions for the binary and the constitutive sources. The distributions generated by the binary source are spread along a large range of values of *n*. The distributions of *n* emitted by the binary source approach that of a constitutive gene as $p_{\alpha }\to 1$.

The graphs of Section 3.3 show that at maximal mutual information and reduced entropy, the distributions governing the message are bimodal. That implies the slow switching and average duration of the ON or OFF states of the promoter being large enough for the number of transcription factors to reach steady state before the next switching. Hence, if we consider the conditional entropy given the promoter is ON, it will behave as a constitutive source. In other words, that regime permits the transcription factors to be a read out of the state of the promoter of the gene. As the parameter ϵ increases, the super-Poissonian distributions change their shapes, e.g., becoming uniform-like, negative-binomial like, or Poissonian-like. Those regimes will have decreasing maximal mutual information and reduced quality on the information content of the message.

## 4. Discussion

The advent of fluorescence techniques generating precise molecular level data has encouraged the use of theoretical approaches on the search for general principles underpinning biological processes. Here we contribute to these efforts by means of an information theoretic analysis of the well-known stochastic binary model for gene expression. The bimodal probability distribution obtained from this model, characterized by large noise, optimizes the mutual information (see Figure 2 or [19,30,31]). Because our results are derived using an exactly solvable master equation modeling a small set of effective reactions, they provide guidance for the interpretation and description of experimental results. Though we are not claiming that we have a model for expression of the *Drosophila* embryo developmental gene *hunchback*, we use its expression as an example. The histogram of the levels of Hunchback protein along the AP axis of a *Drosophila* embryo at cycle 14 is also bimodal, and characterize a near optimal information flow in transcriptional regulation [19]. Expression of *hunchback* is modulated by self-activation, repression by Knirps and Kruppel transcription factors, and activation by the maternal protein Bicoid [53]. In a “first order” approximation, however, one may assume Bicoid as a major regulator of *hunchback* gene such that with some precision [53], its promoter state is a readout of the concentration of Bicoid—low concentration of Bicoid implies on the promoter being OFF and vice-versa [36]. In that case, the bimodality of the distribution of Hunchback suggests that its promoter has a slow switching dynamics and, since the state of the promoter is regulated by multiple transcription factors [54], we assume that their binding and unbinding holds its promoter at either ON and OFF states for a relatively long interval. This incipient analysis is important to be completed because of the similar kinetics of the gap genes despite their distinctive spatial patterns [14]. Furthermore, the number of elongating transcripts in a gene with promoter controlled by distinct developmental enhancers of *Drosophila* exhibit trajectories consistent with the bimodal distributions [22]. Hence, mapping the biological systems in which binary expression occurs may provide useful general principles to understand the functioning of gene networks in the context of development.

A major challenge in Biology is to classify noise in gene expression accordingly with the role of its products in a functional network or a cell. For example, self-repression has long been recognized as the mechanism responsible for noise reduction or homeostasis in gene expression [55,56,57,58]. Our results indicate that in a gene network in which the relevant information is the promoter state, the binary expression is the proper regime to increase the reliability and information content of the message for a wide range of the noise amplitudes. Another noise regime of gene expression is the bursting, which generates a message with minimal entropy and low mutual information (Figure 2). The bursts have been reported in many recent publications, e.g., see Table 1 [59] for a compilation showing average values of burst size and frequency among ∼4k and ∼2k genes of, respectively, mouse and human fibroblasts cells [8,60,61,62,63,64], or two genes in yeast [51]. The recurrence of observations of burstings indicates the necessity of additional analysis using more general models [59,65] to understand the flow of information in gene networks with bursty components. One may consider that the bursty genes are participants of networks in which the relevant information is not the promoter state, and they may even not need to operate at a near optimal regime of promoter state information transmission. These three examples, self-repression, binary and bursty expression, lead us to *conjecture* that evolution of living systems do not necessarily select or avoid selecting gene networks for maximal information flow, but for near optimal operation under the necessity of sufficiently reliable function and near minimal energy consumption.

Information transfer in biological systems also play a role on phenotypic switching for adaptation to constantly changing environments. When the environment fluctuates, two strategies (and their combinations) are possible [40]: (*i*) sensing and response; and (*ii*) blind, random switching. In case (*i*) we a assume a cell which survival in a given environment depends on the presence of a specific protein. The occurrence of the environment leads to the expression of the specific protein by activation of its corresponding promoter. The optimal response regime would require the protein numbers to be a readout of the promoter state and, for a fluctuating environment, the trajectories of the protein numbers would be similar to those described for bimodal distributions (see Section 2.3.3). The duration of the high concentrations of the responding protein would be similar to duration of the demanding environment. Note, however, that the chemicals responsible for activating the responding gene may have a specific production regime optimized for detection of environmental changes. In case (*ii*) there is a random switching that does not respond to the environment [39,41]. How do the cells implement the optimal switching rates? For instance, they could just have a randomly switching gene with arbitrary, optimal ON→OFF and OFF→ON rates. However, the phenotype comes from the protein level, not from the gene. So, if the randomly switching gene’s ON/OFF states are not faithfully followed by the protein, the phenotypic switching will not be optimal, even though the gene switching is optimal. So, for optimal random phenotypic switching, there should be optimal information transfer from the gene’s state to the protein levels.

Our results may also help unravel why transcription factors are present in lower copy numbers inside the cells when compared with, e.g., glycolytic or elongating proteins (see the online book http://book.bionumbers.org/what-are-the-copy-numbers-of-transcription-factors/ related to BioNumbers webpage—[66]. Assume that the transcription factors are the carriers of information about the promoter state and that their interaction with the regulatory regions of the receptor gene depends on their quantity. One may wonder whether there is a threshold above which the amount of transcription factors is an unnecessary burden for the cell because of lack of additional information and their near certainty of binding to the DNA. Indeed, Section 3.3 shows that the mutual information between the message and promoter state saturates for a finite value of the average number of transcription factors while entropy grows monotonically. It would be interesting to connect this result with the functioning of specific gene networks. As previously discussed, gene networks may or may not function transmitting information about promoter states and further analysis is necessary to understand additional possibilities relating a gene network function and its noise regime.

For example, there have been long term characterization of the enhancers as the gene structures responsible for regulating the promoter exposure for transcription and, consequently, average levels of gene products. Assuming that all gene expression noise regimes have a functional role, one may wonder whether there exist a DNA structure surrounding a promoter that regulates its switching rates and, consequently, whether its expression will be binary, bursty, constitutive-like accordingly with its context necessities. That would be interesting to investigate experimentally to relate an expression regime with its functional role. Additionally, one might also test the mutual information between a transcription factor and its target protein, or their promoters. Another interesting question is to test the information transfer between an external signal and genes involved in its processing.

## Figures and Tables

**Figure 1 entropy-22-00479-f001:**
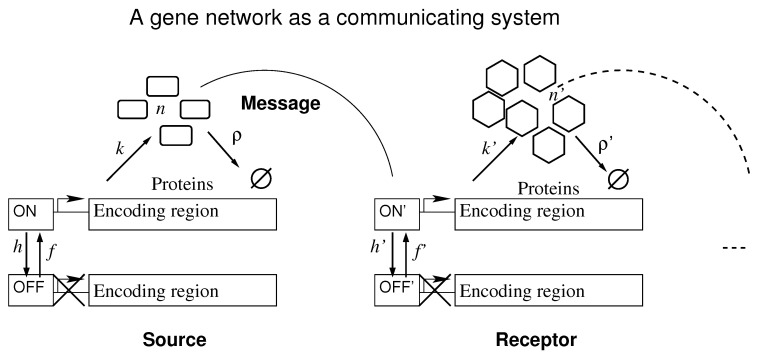
Two elements of an arbitrary gene network are depicted as a communicating system. The continuation of the network is represented by the dashed lines. A source gene communicates with a receptor gene through transcription factors and the information is the promoter state of the source.

**Figure 2 entropy-22-00479-f002:**
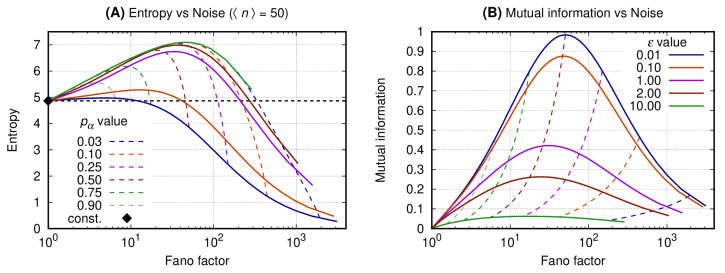
Entropy (or mutual information) and Fano factor were obtained varying *N* for fixed $p_{\alpha }$ and ϵ. All curves have the same value for the average number of gene products (〈n〉=50 such that for each curve we have $N=50/p_{\alpha }$). Graph (**A**) shows the entropy of the message sent by the gene versus the noise of the message. Graph (**B**) shows the mutual information between the message and the source state as a function of the noise. Continuous lines refer to fixed values of ϵ and fixed values of $p_{\alpha }$ are indicated by the dashed ones.

**Figure 3 entropy-22-00479-f003:**
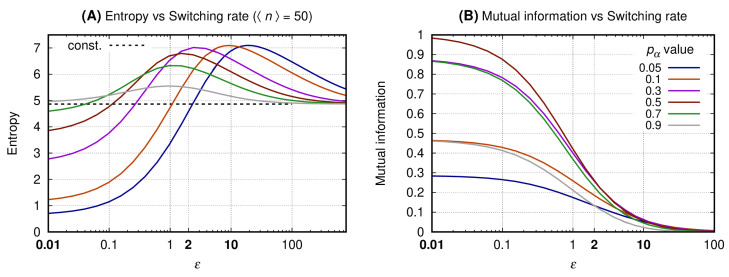
The entropy (and mutual information) was obtained fixing $p_{\alpha }$ and *N*. All curves have the same average number of transcription factors (〈n〉=50 such that for each line we have $N=50/p_{\alpha }$). The horizontal axis of both graphs indicates the transcription factor’s lifetime relative to the source’s ON and OFF promoter switching duration. Graph (**A**) shows the entropy of the message and Graph (**B**) shows the mutual information between the message and the source’s promoter state. Each continuous line corresponds to a given value of $p_{\alpha }$ and the entropy value for a constitutive gene producing 〈n〉=50 proteins is shown by the horizontal dashed black line. Values of ϵ used to construct Figure 4, Figure 5 and Figure 6 appear in bold at the horizontal axis label.

**Figure 4 entropy-22-00479-f004:**
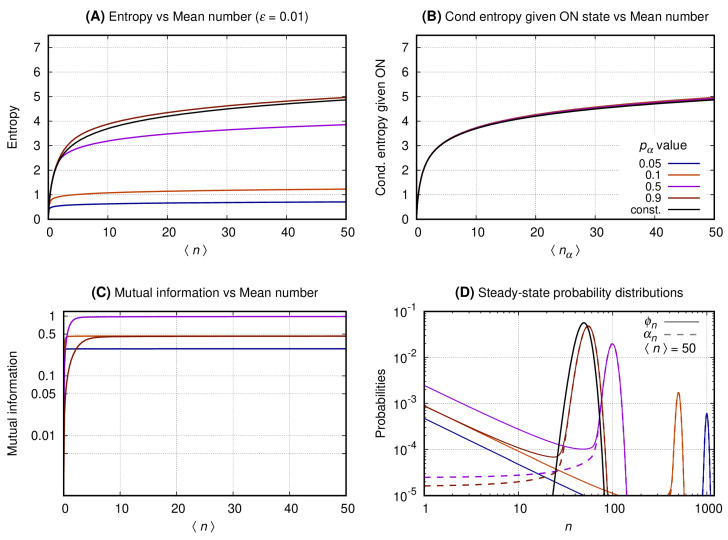
The switching rate of the promoter of the source is 0.01. The vertical axis of Graph’s (**A**–**C**) shows, respectively, entropy versus 〈n〉, entropy conditional to the source being ON versus $\langle n_{\alpha }\rangle$, and the mutual information versus 〈n〉. Graph (**D**) shows the probability distribution of *n*.

**Figure 5 entropy-22-00479-f005:**
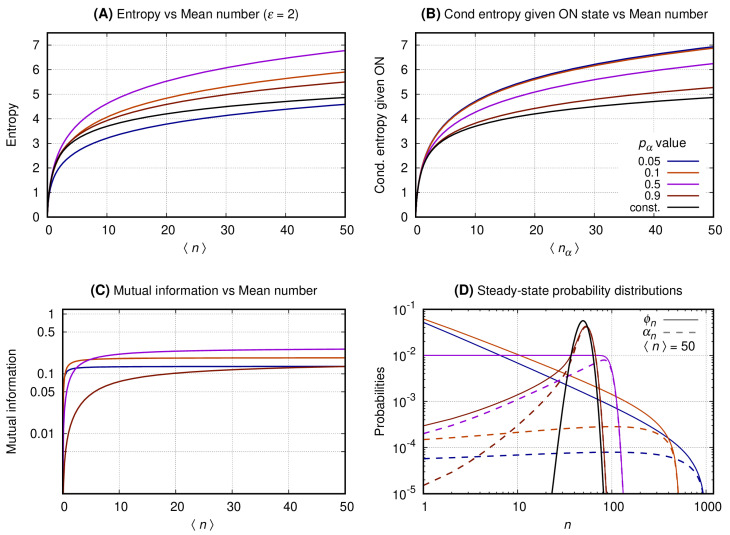
The switching rate of the promoter of the source is 2. The Graph’s (**A**–**C**) shows, respectively, entropy versus 〈n〉, entropy conditional to the source being ON versus $\langle n_{\alpha }\rangle$, and the mutual information versus 〈n〉. Graph (**D**) shows the probability distribution of *n*.

**Figure 6 entropy-22-00479-f006:**
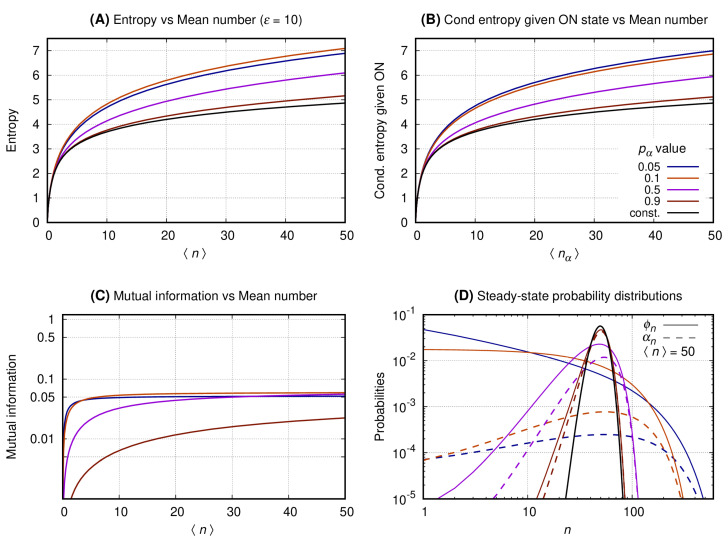
The switching rate of the promoter of the source is 10. The vertical axis of Graph’s (**A**–**C**) shows, respectively, entropy versus 〈n〉, entropy conditional to the source being ON versus $\langle n_{\alpha }\rangle$, and the mutual information versus 〈n〉. Graph (**D**) shows the probability distribution of *n*.

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
