# Peer review of "Binary Expression Enhances Reliability of Messaging in Gene Networks"

_entropy, 2020, doi:10.3390/e22040479_

Round 1

Reviewer 1 Report

The paper by Gama et al discusses how the bursty expression of genes may enhance the reliability of messaging in gene networks. The authors use the well-known exact solution of the two state model of gene expression to quantify Shannon’s entropy of the message (transcription factor) and the mutual information between the message and the promoter state. I enjoyed reading the paper and I think its a nice contribution to the literature. However I believe their interpretation of some of the results is incorrect. In particular I have two concerns:

Minor

In the two state model as used in the literature, the product is is not a transcription factor but rather the messenger RNA (mRNA). See for example Raj, Arjun, et al. "Stochastic mRNA synthesis in mammalian cells." PLoS biology 4.10 (2006) which is one of the main papers that popularized the model. Transcription factors are proteins and hence cannot be seen as the direct output of gene expression but rather as a secondary output after translation from mRNA. Hence I suggest the authors to change the product to mRNA.

Major

The authors define early in the manuscript, bursty expression of a gene when its products are governed by a super Poissonian distribution, i.e. the Fano factor is greater than 1. This is certainly not the definition of burstiness used in the literature of gene expression. The standard definition is that expression is in bursty when a gene spends most of its time in the OFF state and simultaneously when the synthesis rate is high. This implies that the gene is OFF most of the times but when briefly ON, it produces a lot of mRNA and hence the observed bursts. The definition used by the author's will not necessarily give rise to bursts as understood in the biology literature because the Fano factor = 1 + mean burst size and hence is a measure of the mean amount of mRNA produced when the gene is ON but has no information about the fraction of the time that the gene is OFF. Hence the Fano factor could be high but the gene could be mostly ON which of course implies no bursty expression! This point has been made several times in the literature, most notably this was first realized and reported here "Single-RNA counting reveals alternative modes of gene expression in yeast." Nature structural & molecular biology 15.12 (2008): 1263. In particular see the discussion on P. 1267 of this paper. Hence the correct mathematical definition which gives bursty expression is the limit that h and k go to infinity such that h/k is a constant. In this limit the solution of the two state model is well approximated by a negative binomial and there is no bimodality -- thousands of genes in yeast, mouse and human cells show this behaviour -- see Table I and the results of the paper "Analytical distributions for detailed models of stochastic gene expression in eukaryotic cells." Proceedings of the National Academy of Sciences (2020). This re-interpretation of what burstiness is, has major implications for the interpretation of the current paper. The authors say that slow switching with equal times spent in the ON and OFF states reduces entropy and maximizes information but this is not a burstiness regime as explained above. Further more as shown in Table I of the above paper, the fraction of the time spent in the ON state for most genes is very small and the distribution is hence unimodal in most cases. Hence what this shows is that nature is not trying to maximize information, at least not the information considered here. In this reviewer's opinion, this is indeed the result of the current paper. This does not detract from the novelty of the paper. I would still support the paper's publication if it is properly rewritten to change the message and appropriate references as mentioned above used to support their arguments.

Author Response

Response to Reviewer 1

We first would like to thank you for the encouragement and useful comments that helped us to improve of our manuscript. Your comments appear in blue below and our responses are itemized using bold faced numbers. For the text references cited in our response, please see the attachment.

1. Minor concern

In the two state model as used in the literature, the product is is not a transcription factor but rather the messenger RNA (mRNA).

1. In the 4th paragraph (Page 2, lines 58-67) of the Introduction we approach this topic. The former version of the stochastic binary model was applied to model protein synthesis (2nd sentence). We warn the readers that an enhancement of the former version of the binary model is available (3rd sentence). The limitations of our approximation and experimental results justifying it are discussed (4th and 5th sentences).

2. Major concern

2.i The definition used by the author's will not necessarily give rise to bursts as understood in the biology literature because the Fano factor = 1 + mean burst size and hence is a measure of the mean amount of mRNA produced when the gene is ON but has no information about the fraction of the time that the gene is OFF. Hence the Fano factor could be high but the gene could be mostly ON which of course implies no bursty expression!

2.i We were not precise in the manuscript and thank you to bring this topic to our attention! Sub-subsection 2.3.3 (pages 5 and 6, lines 139-179) addresses this by qualitatively distinguishing the noise regimes based on Fano factor values. We renamed the optimal mutual information regime as binary to avoid a confusion with the literature reporting the bursts.

2.ii Further more as shown in Table I of the above paper, the fraction of the time spent in the ON state for most genes is very small and the distribution is hence unimodal in most cases. Hence what this shows is that nature is not trying to maximize information, at least not the information considered here.

2.ii We have reformulated our manuscript and discussed the necessity of a classification of the noise in gene expression accordingly with the function of its products within a network or cell. That is mostly discussed at Section 4 - Discussion (pages 12-14, lines 347-405). Table 1 of Cao2020 makes an average of the ON state probability among ~2k and ~4k genes, representing about 10% of the human and mouse genomes that are composed by about 20k and 30k genes, respectively. Indeed, the obtained ON state probabilities are averaged among genes of either human or mouse fibroblasts suggesting the predominance of bursty expression. However, since: (i) it is an averaging that hides the individual genes' specificities; (ii) the data is biased by the limitations of current single-cell sequencing technologies, that can detect abundances reliably only for the highest-expressed transcripts depending on the maximum read count (depth of sequencing), and by the reliability of the inferences; and (iii) cells of only three species (yeast, mouse, human) are being used; we consider that it is still too early to draw definitive conclusions. Hence, we prefer to initiate and promote a discussion (page 14, lines 379-383) within the research community until sufficient data of appropriate quality accumulates to support a clear interpretation.

Reviewer 2 Report

This is a very interesting research work and can be very useful to elucidate the biochemical mechanisms in a gene network. The idea of considering a gene network as a communicating system can help to understand how the gene regulation works.

However, from my point of view and as a reader, there are some issues that must be solved.

For example, the introduction does not provide enough information about the research problem, it's hypothesis or how other authors have faced the same challenge. You can not refer to seventeen research works in the bibliography in a single line (Line 22). It's very difficult to know if your idea is a new one or an evolution from another. The basis and the justification of your work is not well set out.

The same occurs with the results. You just comment them, but you don't explain how these results have been generated, the different types of tests, the input data that has been used, a comparison with other works....So, i cant figure out the meaning or the relevance of your experimental work.

Finally, the organization of the sections must be improved. It is not easy for a reader to find a reference to a equation in the page number 2 when this equation is explained in the page number 9.

In conclusion, this work has some issues to be solved if someone external to the research work want to understand it.

Author Response

Response to Reviewer 2

We first would like to thank you for the encouragement and very helpful comments, the revised manuscript is better after responding to them. Since we agree with you, we indicate where our responses appear in the manuscript. Your comments appear in blue and our responses next in black, accordingly with the numbers in boldface. For the text references cited in our response, please see the attachment.

1. For example, (i) the introduction does not provide enough information about the research problem, (ii) it's hypothesis or how other authors have faced the same challenge. You can not refer to seventeen research works in the bibliography in a single line (line 22). (iii) It's very difficult to know if your idea is a new one or an evolution from another. The basis and the justification of your work is not well set out.

We have rewritten the Introduction as follows:

1.i we dismantled the citations specifying their content (Paragraphs 1 to 4. Pages 1 and 2, lines 18-67);

1.ii Paragraph 2 (page 2): sentence 4 (lines 38-40) justifies our hypothesis of the information being the promoter state; sentence 5 (lines 40-43) justifies our hypothesis of the message being the number of transcription factors; sentence 6 (lines 43 and 44) sets the scope of the current study.

1.iii Paragraph 3 (page 2, lines 45-57): discusses previous applications of Shannon's theory to understand information flow in biochemical networks. Those studies were parameter-free and formulated in the linear noise regime. Our study was motivated by the phenomenology described in 1.ii, and, with all respect to researchers preceding us on biological applications of information theory, it is a new idea. It enables the addition of a biological and mechanistic interpretation of the results because of the exactly solvable stochastic model for a minimal set of effective reactions governing gene expression. Paragraph 4 (page 2, lines 58-67): discusses the basis for the choice of the stochastic binary model.

2. (i) The same occurs with the results. You just comment them, but you don't explain how these results have been generated, the different types of tests, the input data that has been used, a comparison with other works... (ii) So, i cant figure out the meaning or the relevance of your experimental work.

2.i We started the Results section (page 7, lines 200-210) with the rationale driving our calculations.

2.ii Unfortunately, at this stage we still do not have an experiment and, to the best of our knowledge, this is the first paper on this subject using the stochastic binary model for gene expression. We expect to perform experimental work in the future. At the introduction we discuss some potential experimental systems for which our methods can be useful (2nd paragraph). Additionally, at Section 4 (page 12-14, lines 347-405) a discussion on the potential meaning and relevance of our work for analysis of experimental data is presented.

3. Finally, the organization of the sections must be improved. It is not easy for a reader to find a reference to a equation in the page number 2 when this equation is explained in the page number 9.

3. We rearranged the order of the manuscript and "Model and methods" became Section 2 (page 3-7, lines 79-198). We also include a first paragraph describing the rationale guiding us on the presentation of the methods. A link to the programs used to perform the calculations of the presented quantities is given at the end of the 1st paragraph.

Round 2

Reviewer 1 Report

I thank the authors for a revision that addresses the bulk of my concerns. As I stated in my first report, I like this work and I think it makes an interesting addition to the literature on gene expression. There is one point that I would like the authors to briefly consider.

In the Discussion section, it is written "Some genes in mouse and human fibroblasts are expressed in bursts, etc.." and they reference Larsson et al. (2019). The way this sentence is written reads like bursty expression is typical for a few genes and not necessarily that common. I believe this is not a true representation of the literature. An analysis of each of the genes reported in that paper shows that practically each one of them (and there are thousands) has bursty expression - while this is a small proportion of the total number of genes, nevertheless bursty expression is found independent of the cell type and the gene function, which makes a strong argument for bursty expression being the commonest known expression mode to date in eukaryotic cells. To bolster this argument and to show to the reader more clearly a summary of the data available across many cell types, for thousands of genes in various experimental conditions, I would consider it very helpful if the authors made explicit reference to Table I in Cao and Grima. "Analytical distributions for detailed models of stochastic gene expression in eukaryotic cells." Proceedings of the National Academy of Sciences 117.9 (2020): 4682-4692. While the data in this table is taken from many publications, to my knowledge it is the only paper which brings together the results from several publications to make a strong convincing case for the ubiquity of bursty gene expression in eukaryotic cells. I believe this is an important addition for two reasons: (i) in this regime of bursty expression, information is not maximal; (ii) the authors discuss application of their model to embryo developmental gene expression which is well known to be bursty (Ref 14). Besides the Cao et al. publication provides the latest and most detailed extension of the telegraph model (used in this paper) to account for various salient features of cellular biology and hence would make an excellent candidate for future studies investigating the mutual information between the message and the promoter state; something that can be mentioned in the Discussion as well. 

Author Response

Thank you for detailed considerations of the bursts and the summary presented at Table 1 of Cao and Grima (2020). We have discussed it explicitly at pages 13 and 14 in the blue sentences. 

Reviewer 2 Report

All the issues detected have been successfully fixed. So, the paper has improved clearly.

Author Response

Thank you for your comments! We acknowledge them in the manuscript.